# The Impact of Alternative Fuels on Ship Engine Emissions and Aftertreatment Systems: A Review

**Shuo Feng [1,†], Shirui Xu [1,†], Peng Yuan [1,2], Yuye Xing [1], Boxiong Shen [1,2,*], Zhaoming Li [1], Chenguang Zhang [1], Xiaoqi Wang [1], Zhuozhi Wang [1], Jiao Ma [1] and Wenwen Kong [1]**

[1] Tianjin Key Laboratory of Clean Energy and Pollution Control, School of Energy and Environmental Engineering, Hebei University of Technology, Tianjin 300401, China; fengshuo0001666@163.com (S.F.); xuthree2022@163.com (S.X.); yuanpeng@hebut.edu.cn (P.Y.); xyy123120101@163.com (Y.X.); m13283202813@163.com (Z.L.); 2287588410a@gmail.com (C.Z.); xiaoqi914879173@126.com (X.W.); windowsxxpp@126.com (Z.W.); majiao2019@hebut.edu.cn (J.M.); 2019914@hebut.edu.cn (W.K.)

[2] School of Chemical Engineering, Hebei University of Technology, Tianjin 300130, China

\* Correspondence: shenbx@hebut.edu.cn

† Co-first author, these authors contributed equally to this work.

**Abstract:** Marine engines often use diesel as an alternative fuel to improve the economy. In recent years, waste oil, biodiesel and alcohol fuel are the most famous research directions among the alternative fuels for diesel. With the rapid development of the shipping industry, the air of coastal areas is becoming increasingly polluted. It is now necessary to reduce the emission of marine engines to meet the strict emission regulations. There are many types of alternative fuels for diesel oil and the difference of the fuel may interfere with the engine emissions; however, PM, HC, CO and other emissions will have a negative impact on SCR catalyst. This paper reviews the alternative fuels such as alcohols, waste oils, biodiesel made from vegetable oil and animal oil, and then summarizes and analyzes the influence of different alternative fuels on engine emissions and pollutant formation mechanism. In addition, this paper also summarizes the methods that can effectively reduce the emissions of marine engines; it can provide a reference for the study of diesel alternative fuel and the reduction of marine engine emissions.

**Keywords:** alternative fuel; after treatment; diesel; emission; ship

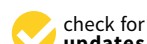



## 1. Introduction

The rapid development of society has not only improved people's living standards, but also enhanced their awareness of environmental protection. China has a vast sea area and inland rivers, which could bring great economic benefits; however, ship emissions are constantly harming the air environment of coastal cities. The pollutants emitted from ships mainly include solid particles (10–100 nm in diameter) and gaseous pollutants, such as $NO_X$, HC (hydrocarbon), CO, and $SO_2$. The conceptual model of the combustion process for diesel engines and the formation process of $NO_X$ is shown in Figure 1 [1], and the chemical reactions equations of formation for thermal $NO_X$ were expressed by Equations (1)–(5) [2–4]. Moreover, the generation process and the structure of soot were presented in Figure 2 [5]. In addition, the emissions of HC or CO were attributed to the incomplete combustion of fuel [3,6].

$$O + N_2 = NO + N \tag{1}$$

$$N + O_2 = NO + O \tag{2}$$

$$N + OH = NO + H \tag{3}$$

$$NO + HO_2 = NO_2 + OH \tag{4}$$

$$NO_2 + O = NO + O_2 \tag{5}$$

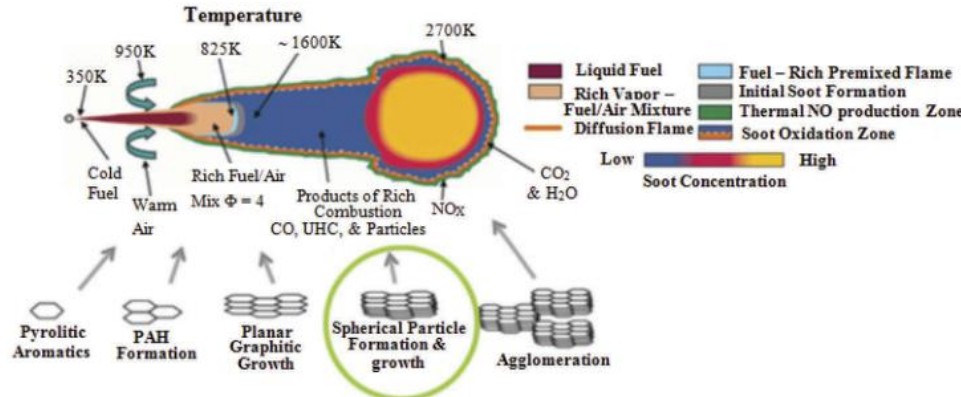

**Figure 1.** Combustion process for diesel engine and formation process of soot and $NO_X$. Reprinted with permission from [1]. Copyright © 2022, Taylor and Francis Ltd.

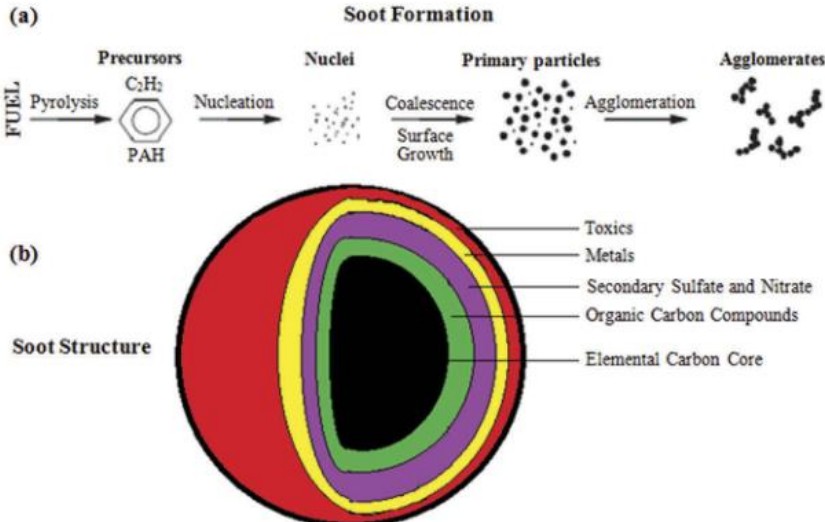

**Figure 2.** The generation process of soot (**a**); the structure of soot (**b**). Reprinted with permission from [5]. Copyright © 2022, Elsevier B.V.

At the same time, a series of pollutants, such as oil leakage from ships, have been released [7]. The International Maritime Organization (IMO) has made strict regulations on the emission of the marine diesel engine. In order to reduce the harm of engine emissions to the human body, the United States, Sweden, Japan, Russia and other countries have established corresponding jurisdiction institutions and systems. The number of shipbuilding in China is far more than that of other countries in the world. At present, China has more than 800,000 ships of various sizes. According to statistics, China's annual $CO_2$ emissions from ocean-going ships can reach 40 million tons, and $NO_X$ emissions also account for one-third of the world's total emissions [8]. The $NO_X$ of marine engine mainly consists of three types: prompt $NO_X$, thermal $NO_X$ and fuel $NO_X$ [9]. Thermodynamic $NO_X$ is generated under the condition of high temperature and rich oxygen. When the cylinder temperature is high, $N_2$ and $O_2$ react to form $NO_X$. Marine engine has the characteristics of large load, high temperature and high oxygen concentration, leading to the thermal $NO_X$ being the main way to form $NO_X$ in the exhaust of marine engines. $NO_X$ will cause great damage to human respiratory, heart and kidney system. In addition, a large amount of $NO_X$ emission will cause photochemical smog [10–12]. Due to the long service life of Chinese ships, the serious aging of the post-treatment system and other problems, the emission of engine pollutants continues to rise. Therefore, how to reduce ship emissions has become an urgent problem to be solved in China. In recent years, China has paid more and more attention to the emission of $NO_X$, and formulated a series of emission regulations. At present, SCR

technology is considered to be the most effective method to remove $NO_X$ from marine engine exhaust, and the reaction of SCR catalysts are shown in Equations (6)–(8), which the $NO_X$ was reduced to $N_2$.

$$4NO + 4NH_3 + O_2 = 4N_2 + 6H_2O \quad \text{[Standard SCR reaction]} \tag{6}$$

$$NO + NO_2 + 2NH_3 = 2N_2 + 3H_2O \quad \text{[Fast SCR reaction]} \tag{7}$$

$$6NO_2 + 8NH_3 = 7N_2 + 12H_2O \quad \text{[Slow SCR reaction]} \tag{8}$$

Since the exhaust components of marine engines are complex, including $SO_2$, HC, PM and so on, these components will lead to SCR catalyst blockage or poisoning deactivation. Moreover, the high exhaust temperature and the exhaust flow rate have a certain effect on the catalyst. Therefore, in order to reduce $NO_X$ emission from ships, it is necessary not only to improve the working stability and anti-toxicity of SCR system, but also to reduce the generation of $SO_2$, PM, HC and other substances in the exhaust; however, due to the rapid consumption of fossil fuels, marine engines tend to use diesel as an alternative fuel to improve economy, which would lead a more complex compositions in marine engine exhaust. Marine engines have strong applicability to different alternative fuels and can be used directly with a small modification (fuel injection pressure, injection timing and intake pipeline etc.) when the alternative fuels mixed with diesel oil [13–17]. The existing diesel alternative fuels mainly include waste oil, biodiesel, and alcohol fuel. The raw materials were processed and mixed in different proportions to produce alternative fuels for marine engines; however, the composition of different alternative fuels is different, which has a great influence on the exhaust composition of the engine. Therefore, studying the influence law of alternative fuel types on marine engine emissions is helpful to reduce the adverse effect of marine exhaust gas on SCR catalyst, so as to reduce $NO_X$ emissions from marine engine. Issa et al. [18] reviewed the law of pollutant emissions and emission reduction technologies of diesel engines. Moreover, the cost of different emission reduction technologies was analyzed from economic and environmental perspectives. In this study, the alternative fuels were classified and introduced, but the specific properties of the fuels were not summarized. Lu et al. [13] reviewed the principles and research progress of different $NO_X$ emission reduction technologies, and compared the advantages and disadvantages of different $NO_X$ emission reduction technologies. Furthermore, Lu et al. proposed that the natural gas would be an important alternative fuel for future marine engines. Deng et al. [19] analyzed the emission laws of marine engines, and provided the emission reduction ideas of future marine engines from the perspectives of optimized combustion control, after treatment system, and fuel optimization (fuel emulsification, fuel additive and fuel desulfurization). Additionally, Deng et al. proposed the prospect of using natural gas as an alternative fuel for marine engines, which included environmental protection, energy structure and economic benefits. Elgohary et al. [20] reviewed the alternative fuels of marine engines and analyzed their applicability of alternative fuels for marine engines. In addition, Elgohary et al. emphasis on the potential of liquefied natural gas (LNG) as future marine fuel, and compared the different properties of LNG and HFO; however, these studies did not summarize relevant information about other alternative fuels, such as extraction methods, sources of feedstocks, and influence mechanism on marine engine emissions.

Therefore, in this paper, the effects of diesel alternative fuels, such as waste oil, biodiesel, and alcohol fuels on engine emissions are summarized. The toxicity mechanism of alternative fuel emission to SCR catalyst was discussed. In addition, some technologies to reduce marine engine emissions are proposed. At the same time, the prospect of reducing engine emissions and improving the performance of alternative fuels in the future is presented.

However, the alternative fuels need to meet some requirements, which are shown as following: (1) High heating value (The heating value of diesel is 42–44.8 MJ/kg [15,17,21,22], marine diesel ISO 8217 is 42 MJ/kg and heavy fuel oil ISO 8217 is 40 MJ/kg [23,24]).

(2) Greenhouse gas (GHG) neutral. (3) High stability (Easy to transport). (4) Low sulfur content (<0.5% [25–27]).

Therefore, some studies evaluate the existing alternative fuels in terms of energy density, GHG emissions and energy cost etc., and the reasons were shown as Table 1.

**Table 1.** Status of viability for different alternative fuels [19,20,28,29].

| Criteria | HFO | LSFO | LNG | Biodiesel | Methnol |
|---|---|---|---|---|---|
| Energy density | A | A | B | A | B |
| Technological maturity | B | B | B | A | C |
| Local emissions | D | D | B | D | B |
| GHG emissions | E | E | C | B | C |
| Energy cost | A | B | A | D | C |
| Capital cost | A | A | B | A | B |
| Converter storage | B | A | C | A | B |
| Bunkering availability | A | A | B | D | C |
| Commercial readiness | A | A | A | C | B |
| Flammability | A | A | A | A | C |
| Toxicity | A | A | A | A | C |
| Regulations and guidelines | A | A | A | A | B |
| Global production capacity and locations | A | A | A | D | B |
| Renewability | D | D | D | C | B |
| Safety | A | A | A | A | B |

A–E: status rating with E being extremely poor and A being excellent; HFO: Heavy fuel oil; LSFO: Low suphur fuels; LNG: Liquefied natural gas.

## 2. Waste Oil

At present, there is a shortage of non-renewable resources, such as fossil fuels [23,30]. One of the ways to solve this problem is to recycle waste oil as an alternative fuel for diesel [31]. It is found that waste oil is a rich alternative resource for diesel oil, with an annual global output of 24 million tons [32]. Waste plastics, waste lubricating oil and waste edible oil are all available waste oil resources [33]. It has become a research direction in recent years to mix waste oil with commercial diesel for marine engines. In addition, compared to other waste oil treatment management methods, burning it as an alternative fuel is an ecologically sustainable, socially admissible, cost-effective solution arises [34,35].

### 2.1. Waste Plastic Oil

The waste plastic oil (WPO) produced by pyrolysis can effectively help solve the problems of diesel shortage and waste plastic treatment [36], while waste plastic oil can replace diesel as engine fuel without major modifications to the engine [37]. The production process diagram of waste plastic oil is shown in Figure 3 below [38]. Firstly, different kinds of waste plastics were cleaned to remove dirt and waste, then the mixed waste plastics were heated and melted, and then the waste plastic oil was obtained by condensation and reheating.

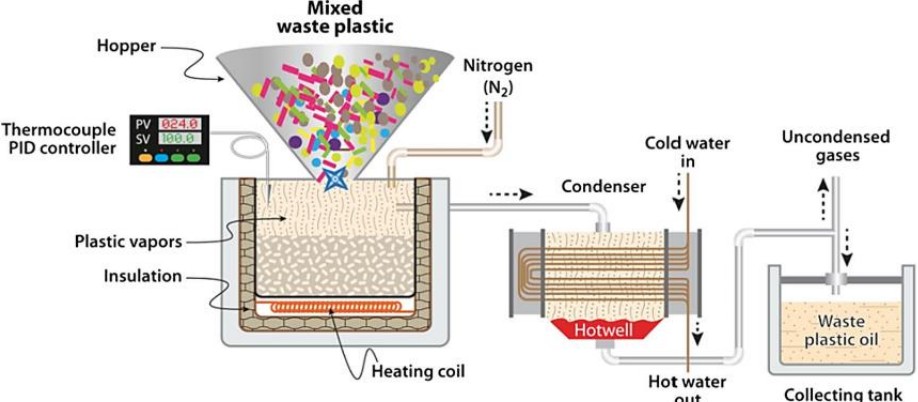

**Figure 3.** Production diagram of waste plastic oil. Reprinted with permission from [38]. Copyright © 2022, Elsevier B.V.

Mani et al. [39] found that waste plastic oil fuels increase $NO_X$ emissions from engines, mainly because waste plastic oil fuels have a higher heat release rate and a higher combustion temperature, which is conducive to the reaction of $N_2$ and $O_2$ to form $NO_X$ under high temperature and pressure, thus increasing $NO_X$ emissions. The HC emission of diesel oil and waste plastic oil is basically the same at low load, and the HC emission of waste plastic oil is slightly higher than that of diesel oil at high load. Rinaldini et al. [36] prepared waste plastic oil and studied the influence of waste plastic oil on engine power and emissions through comparative experiments. It was found that compared with commercial diesel, the addition of waste plastic oil could reduce PM emission, and then reduce the toxic effect of PM on SCR catalyst. Through further research, this is mainly because the waste plastic oil is rich in volatile hydrocarbons, which can improve the fuel evaporation rate and reduce the carbon deposition of unburned fuel. Moreover, the waste plastic fuel contains oxygen, which can make the fuel burn fully during combustion, thus reducing the generation of PM. Devaraj et al. [37] used waste plastic oil mixed with ether as fuel for diesel, and studied its emission characteristics. The experimental results show that the addition of ether to waste plastic pyrolysis oil will increase HC emission. This is mainly due to the addition of ether, which will reduce the viscosity of the fuel, so that the fuel will leak through the nozzle [40]. The poor mixture of leaked fuel and oxygen will lead to incomplete burning, and thus will cause an increase in HC emission. Some ether will enter the gap between piston and cylinder when the fuel is injected, so the flame surface cannot be touched, which also leads to the increase in HC emission; however, the PM emission of the fuels decreased, which added the ether. With the increase of ether content, PM emission decreased more obviously; this is mainly because the oxygen-containing functional groups in ether can participate in the oxidation of fuel together with oxygen, which can improve the air-fuel ratio of combustion and help to reduce the PM emission of engines.

### 2.2. Waste Lubricating Oil

In recent years, it has been found that waste lubricating oil is an important source of fuel [41]. According to the surveys, about 50% of the fresh lubricants consumed will become waste lubricant oil every year all over the world [42]. Additionally, nearly 780 million tons of lubricating oil are wasted every year in China [43]. Moreover, there are about 750 million liters lubricating oil become waste per year in Japan, which comes from automobiles and marine engines [44]. In addition, El-Mekkawi et al. believe that using waste lubricating oil as an alternative fuel is an environmentally friendly and economical solution [45]. Waste lubricating oil is mainly composed of base oil and additives. The properties of waste lubricating oil are mainly related to the properties of base oil, and additives only play an improvement role. Waste lubricating oil can be regenerated by chemical removal of heavy metals, and can be used as marine engine fuel after being mixed with ordinary fuel in an appropriate proportion [46]. Gorka et al. [31] studied the effect of waste lubricating oil on engine performance and emissions. It is found that waste lubricating oil is suitable for medium speed marine engine, and using waste lubricating oil as engine fuel can help to reduce $NO_X$ emission. This is mainly because the waste lubricating oil fuel can reduce the premixing strength of fuel and air, which leads to the decrease of the maximum engine temperature and effectively reduces the generation of thermal $NO_X$. Orhan et al. [47] studied the influence of diesel like fuel (DLF) produced from waste engine oil on engine emissions, and the system diagram of producing DLF is shown in Figure 4. Firstly, the waste lubricating oil is filtered, and then it enters the reactor through the selenoid valve. The oil from the reactor is cooled by the condenser, and finally the fuel-like oil is obtained; 60 cc diesel fuel can be extracted from 100 cc waste lubricating oil. By extracting DLF in this way, the fuel cost can be greatly reduced, which is beneficial to improve the economy of DLF. Moreover, Orhan et al. calculated the production cost of DLF and found that the cost is 1.016 Turkish Lira/L, which is a market-competitive price. In addition, the extracted diesel like fuel can be directly used in diesel engines, and the engine performance is good, except for sulfur content, the content of other emissions is low.

In order to reduce the sulfur content in DLF, Orhan et al. [48] treated the diesel fuel (DLF) by oxidation desulfurization (ODS) at 50 °C. The results show that the sulfur content in DLF is reduced from 3500 ppm to 420 ppm, and the low sulfur diesel (lsdlf) is obtained. The effects of low sulfur diesel and commercial diesel on engine performance and emission were studied by comparative experiments. The performance parameters of commercial diesel engines are close to those of low sulfur diesel engines, while the emission of low sulfur diesel engines ($NO_X$, $SO_2$) is relatively small, which is conducive to reducing the toxic effect of $SO_2$ on SCR catalyst and reducing $NO_X$ emission of engines.

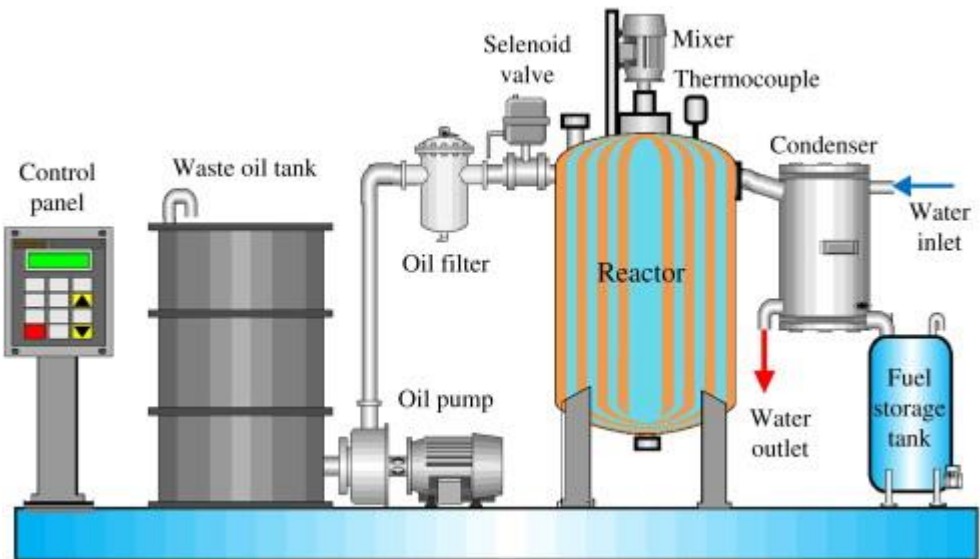

**Figure 4.** Process flow chart for preparation of diesel like fuel from purified waste lubricating oil. Reprinted with permission from [47]. Copyright © 2022, Elsevier B.V.

In addition, Wang et al. [49] also studied the effect of DLF on engine emissions. Compared with diesel, the HC emissions of DLF are higher, which is mainly due to the higher viscosity of DLF, which affects the diffusion of fuel in the engine cylinder, resulting in the increase of unburned HC [50]. Compared with diesel, the PM emission of DLF fuel is higher at medium load. With the increase of engine load, the PM emission also increases. Sacha et al. [51] also prepared a new diesel alternative fuel by CO pyrolysis of waste plastics and waste lubricating oil.

## 3. Biodiesel

Biodiesel is composed of fatty acid methyl ester or ethyl ester, mainly from unprocessed or used vegetable oil or animal fat [52]. Due to the high cost of traditional biodiesel preparation, it is not competitive with commercial diesel. Therefore, in order to reduce the production cost of biodiesel, the research of biodiesel now mainly focuses on the preparation of biodiesel from waste edible oil [14,53–55] and biodiesel from sewage [56]. It is found that the sewage sludge in the city contains a lot of lipids, and using it to produce the biodiesel can greatly reduce the cost [57]. Biodiesel has many natural advantages, such as better lubricity, lower sulfide and aromatic hydrocarbon content, renewable and non-toxic. In addition, there exists oxygen element in the fuel composition, which will help the fuel burn completely, thus reducing the engine emissions [55]. In addition, biodiesel can be directly used in diesel engines without retrofitting the engine structure [14]. It has been found that using biodiesel as engine fuel can effectively reduce the emissions of $SO_2$, HC, PM, CO and other pollutants, but it will increase the emissions of $NO_X$; indeed, using biodiesel as fuel has some disadvantages as well. Because of its high viscosity and low volatility, it cannot form even combustible gas at low temperatures. Moreover, saturated compounds in biodiesel tends to crystallize at low temperature, which leads to the insufficient combustion

during cold start-up. How to solve the problems of high viscosity and low volatility of biodiesel at low temperature is the development direction of biodiesel in the future.

### 3.1. Vegetable Oil

Vegetable oil has a broad prospect as an alternative fuel. As early as 1911, Rudolf Diesel began to study vegetable oil as an alternative fuel [58]. As a renewable energy, vegetable oil has another advantage that it can be produced by many kinds of raw materials, including cotton, mustard, flaxseed, soybean, peanut, sunflower, rape and coconut, etc. [59]. The cetane number and calorific value of vegetable oil are similar to that of commercial diesel oil. Vegetable oil basically does not contain sulfur, and the $SO_2$ concentration in engine exhaust mainly depends on the sulfur content in the fuel [32], so using it as a fuel can help reduce the $SO_2$ content in the exhaust. Vegetable oil has high viscosity and can provide high-quality lubrication for various parts of a diesel engine [59]. Senthilkumar et al. [60] researched the effect of biodiesel on engine emissions, which was prepared from vegetable oil. The report indicates that the use of biodiesel reduces CO, HC and soot emissions of the engine. This is due to the presence of bound oxygen in biodiesel, which is conducive to the oxidation of fuel, thus reducing the emission of CO, HC and soot. But the use of biodiesel can cause the engine's in-cylinder temperature to be too high, which increases $NO_X$ emissions. This is mainly due to the high content of oxygen in biodiesel, which can make the fuel fully burned and help reduce CO, HC and PM emissions.

### 3.2. Waste Edible Oil

Ordinary pure vegetable oil has high cost and is not suitable to use as fuel. In contrast, the price of waste edible oil is one-third to one-half of that of pure vegetable oil [61]. Using waste edible oil as fuel is conducive to solve the current problem of waste edible oil treatment. Waste edible oil is composed of hydrocarbon mixture, including saturated and unsaturated fatty acids, polycyclic aromatic hydrocarbons and organic impurities [61]. Waste edible oil can be converted into diesel fuel by transesterification, which not only reduces the environmental pollution of waste edible oil, but also reduces the energy consumption [62]. Yesilyurt [55] studied the effect of mixing waste edible oil with mixed diesel on engine emissions. Under full load, he studied the emission law of mixed fuel under different working conditions by adjusting engine speed and fuel injection pressure. It is found that the incorporation of biomass diesel reduces the emissions of CO, HC and PM, but increases the exhaust temperature and $NO_X$ emissions. This is because biomass diesel contains oxygen, which will increase the generation of thermal $NO_X$ under the condition of high temperature oxygen enrichment. It can be seen from the data in this paper that $NO_X$ is mostly generated at low speed, because nitrogen and oxygen can have more reaction time at high temperature under low speed, which leads to the increase of $NO_X$ emission. The increase of exhaust temperature and $NO_X$ emission will increase the consumption of SCR catalyst, which is not conducive to reducing engine $NO_X$ emission. Wei et al. [53] studied the emission characteristics of waste edible oil under the 13 mode test cycle in Japan. It is found that the use of biodiesel reduces the concentration and particle size of PM in exhaust gas, which is mainly due to the increase of oxidation of fuel and the reduction of incomplete combustion of fuel. The use of biodiesel increases the formation of formaldehyde, acetaldehyde, 1,3-butadiene, propylene, ethylene and benzene, which is mainly due to the accelerated pyrolysis of long-chain molecules at high temperature and high pressure; however, the addition of biodiesel reduces the emissions of toluene and xylene, mainly because the combined oxygen in the fuel increases the oxidation of benzene ring, thus reducing the emissions of benzene ring.

## 4. Alcohol Fuels

Higher alcohols have better blending ability, hydrophobicity, cetane number, and calorific value, making them excellent alternative fuels for diesel [63]. However, alcohols cannot be directly applied to diesel engines because of their high spontaneous combustion

temperature and high latent heat of vaporization [64], therefore, people generally mix alcohol fuel with other fuels in a certain proportion to supply the engines in order to reduce the $NO_X$ emission. The carbon content of alcohols plays an important role in whether they can be better integrated into diesel or biodiesel. Alcohols with less carbon content, such as ethanol, will cause phase separation at low temperature, and due to its low cetane number, it will lead to low calorific value and poor lubricity of fuel, which will reduce the efficiency of engine and increase the wear of fuel injection nozzle and other parts [65]; however, with the increase of carbon content in alcohols, the compatibility between alcohols and diesel increases, and the oxygen content of alcohols decreases at the same time, which is conducive to the improvement of cetane number and fuel calorific value. In recent years, high carbon alcohol doped fuel has become the focus of research. At present, alcohol fuel is mainly mixed with biodiesel prepared from waste oil or commercial diesel to form engine fuel [63,66,67].

Nadir et al. [64] found that the mixing of ethanol and diesel will reduce the $NO_X$ emission of the engine, but increase the HC and CO emissions, and increase the calorific value of the exhaust. This may be because ethanol has high natural temperature and high latent heat of vaporization [68], resulting in incomplete volatilization of ethanol in the cylinder. Therefore, the combustion of mixed fuel in the cylinder of diesel engine is incomplete, resulting in the reduction of the maximum temperature in the cylinder and the reduction of $NO_X$ emission; however, the volatile ethanol gas in the exhaust pipeline reacts with oxygen again, resulting in the temperature rise of the exhaust system. The increase of HC content in exhaust gas will lead to the blockage of SCR catalyst and even the formation of carbon deposition, resulting in the decrease of catalyst activity. The increase of exhaust steam temperature will lead to the increase of catalyst heat load, accelerate the aging of catalyst, and may increase the risk of sintering. Morsy et al. [69] also studied the effect of the addition of aqueous ethanol and absolute ethanol to diesel on engine emissions. It was found that when the fuel is mixed with aqueous ethanol, $NO_X$ emission is reduced; this is because water has a high specific heat capacity of water, so it absorbs a lot of heat during gasification, resulting in the reduction of cylinder temperature, so as to reduce the generation of thermal $NO_X$. In addition, due to the endothermic of water, ethanol cannot be completely burned; therefore, compared with absolute ethanol, mixing aqueous ethanol with diesel will also increase the emissions of CO and HC. When the fuel is mixed with absolute ethanol, $NO_X$ emission increases; this is because the existence of ethanol will increase oxygen utilization and promote combustion, resulting in an increase in-cylinder temperature, thus promoting the generation of more thermal $NO_X$. Sharbuddin et al. [70] studied the emission of diesel engine equipped with CRDI system using waste edible oil fuel, and also studied the effect of adding $C_8$ and other oxygenates into the fuel on engine emission. It was found that $NO_X$ emission will increase with the increase in oxygen content. When *n*-octanol is added, $NO_X$ emission is the largest. The addition of oxygenates has a significant effect on reducing PM emission, and octanol mixture can reduce PM emission by 75%. Because *n*-octanol increases the oxygen content in the fuel, it makes the fuel burn fully and reduces the emissions of HC and CO.

Alpaslan [63] experimentally studied the effects of higher alcohols such as propanol, *n*-butanol and 1-pentanol mixed with biodiesel prepared from diesel and waste oil on engine power and engine emissions. The mixed fuels of propanol, *n*-butanol and 1-pentanol were prepared by adding 20% volume fraction of propanol, *n*-butanol and 1-pentanol into the mixed fuel of biodiesel prepared from diesel and waste oil, and compared with the fuel without alcohol. It is found that alcohols will lead to the increase of CO emission under low load. Moreover, under low load, alcohol mixed fuels will increase HC emission, except *n*-butanol mixed fuel. The increase of alcohols will significantly reduce $NO_X$ emission under different loads, which is conducive to reducing the working load of SCR catalyst. Alpaslan [66] prepared a microemulsion mixture of hazelnut oil and diesel oil with *n*-butanol/1-amyl alcohol, and added 2-ethylhexyl nitrate to the blended fuel to increase the 16 alkane value of the fuel. It is found that adding 2-ethylhexyl nitrate to the mixture will

improve the combustion characteristics of the mixed fuel and reduce NO$_X$ emission, but it will also increase the emissions of HC and CO. Adding 2-ethylhexyl nitrate will reduce NO$_X$ and HC emissions, but increase CO emissions at the same time. Due to the inherent fuel improvement characteristics of higher alcohols, they can be used as an improver for the mixed fuel of diesel and biomass diesel. As peanut is a relatively cheap oil producing crop, it can be prepared into biomass fuel by technical means. Peanut is widely distributed all over the world, which is conducive to fuel extraction. Peanut planting and yield distribution all over the world are shown in Figure 5. Yesilyurt et al. [67] prepared biodiesel from peanuts by transesterification, and mixed the prepared biodiesel with *n*-heptanol and commercial diesel, in order to study the impact of biodiesel and higher alcohol mixed fuel on the emission of diesel engines, and he also compared it with traditional diesel and pure biodiesel. Compared with biodiesel prepared from pure peanut oil, the addition of *n*-heptane will increase the content of HC in exhaust gas. This is mainly due to the low cetane number and poor spontaneous combustion of *n*-heptane, resulting in incomplete fuel combustion and increased HC emission. Compared with pure biodiesel, the addition of *n*-heptane will reduce NO$_X$ emission, which is mainly due to the high evaporation latent heat of the mixture, which reduces the maximum temperature in the cylinder and weakens the redox reaction between nitrogen and oxygen, thus reducing the generation of NO$_X$.

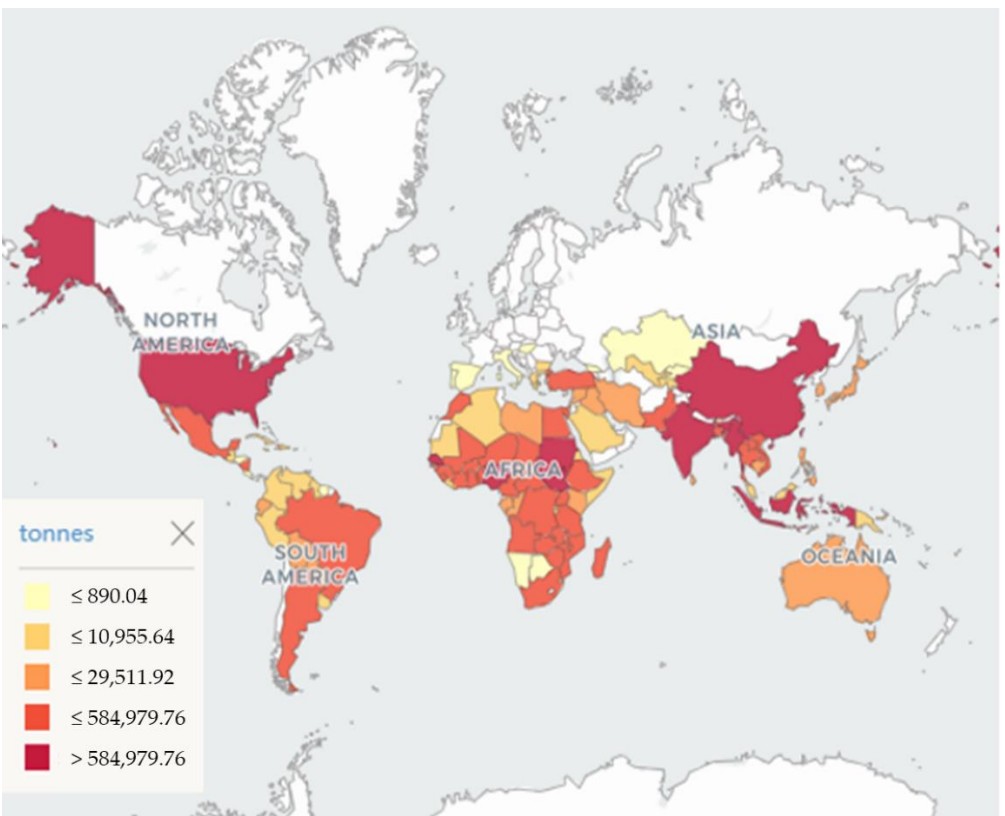

**Figure 5.** World peanut planting and yield distribution. Reprinted with permission from [67]. Copyright © 2022, Elsevier B.V.

Ashok et al. [71] mixed *n*-octanol with calophyllum inophyllum biodiesel in different concentrations to prepare fuel, and tested the effect of mixed fuel on engine emission. it is found that the addition of *n*-octanol will increase the emissions of CO and PM, which is caused by incomplete fuel combustion due to the poor ignition and evaporation of *n*-octanol. With the increase of *n*-octanol concentration, the emissions of CO and PM will increase accordingly. *N*-octanol has high latent heat of evaporation. With the addition of *n*-octanol, the maximum combustion temperature in the cylinder will be reduced, the oxidation of nitrogen will be weakened, and NO$_X$ emission will be reduced. The decrease

of temperature will also reduce the homogeneous and heterogeneous oxidation of CO and PM by $NO_X$, resulting in the increase of CO and PM emissions. Cheung et al. [72] mixed methanol with biodiesel as engine fuel to study the effect of methanol biodiesel mixture on engine emissions. It is found that compared with biodiesel, the addition of methanol will increase the emissions of CO and HC, while the emissions of PM will decrease slightly. This is mainly due to the large latent heat of methanol vaporization and more volatile heat absorption, resulting in low cylinder temperature and incomplete fuel combustion under low load conditions, resulting in increased CO and HC emissions. Through experiments, Babu et al. [22] found that under 100% load, when 10% pentanol is mixed with 85% biodiesel or 5% diesel as fuel, it has the lowest CO and HC emissions. Compared with pure diesel fuel, CO and HC emissions decreased 0.36% and 39.4%, respectively.

## 5. Natural Gas

Natural gas is a kind of diesel alternative fuel with rich reserves and low price. Using natural gas as an alternative fuel for diesel engine can reduce $NO_X$ emission. Since it does not contain sulfur elements, using natural gas as fuel can also reduce $SO_X$ emissions. Sulfur can also promote the formation of PM, so the use of natural gas fuel can also reduce the PM emission of the engine [73,74].

The main component of natural gas is methane, and due to climate, geographical location and other factors, natural gas also contains carbon dioxide, helium and other components. One of the advantages of natural gas as an alternative fuel for diesel is that natural gas is abundant and widely distributed. Different from alternative fuels, such as biodiesel and methanol, natural gas is widely distributed in China, while fuels concentrated in some regions can only be mainly used locally. Over the past two decades, the recoverable reserves of natural gas have increased by about 50%; by 2016, there were about 186 trillion cubic meters. At present, natural gas is mainly used in heating, power generation etc., and it also plays a certain role in preparing chemical raw materials and used as fuel. In addition, as an efficient and clean energy, natural gas can also replace sulfur-containing heavy fuel oil. Since natural gas does not contain sulfur elements, using natural gas as fuel can reduce the emission of sulfur oxides. The presence of sulfur will promote the formation of PM, so natural gas as fuel can reduce the PM emission of engine. In addition, natural gas is mainly composed of simple hydrocarbons, which will also lead to less PM emissions as well [75]. It is found that compared with diesel engines, most natural gas engines produce lower $NO_X$ emissions. The main reason is that natural gas engines use lean combustion and the cylinder temperature is low then, so as to curb the formation of thermal $NO_X$ [75]. At present, the fuels containing natural gas used in practical application mainly include pure natural gas, natural gas gasoline mixture and natural gas diesel mixture [76].

At present, natural gas has been applied to transportation. In China, about 5 million natural gas vehicles have been put into use. Natural gas gasoline blends are often used in taxis and a small number of private cars [77]. Jahirul et al. [78] transformed the 1.6 L 4-cylinder gasoline engine into a natural gas gasoline dual fuel engine, and compared and analyzed the emission characteristics of the two systems. It is found that when the valve opening is 80% and the rotating speed is 1500–5500 rpm compared with gasoline fuel, the addition of compressed natural gas significantly reduces the emissions of HC and $CO_2$, but the emission of $NO_X$ increases by about 40.84%. Aslam et al. [79] also found through experiments that compared with gasoline, in the emissions of dual fuel engine, CO is reduced by 80%, $CO_2$ is reduced by 20%, HC is reduced by 50% and $NO_X$ emission is increased by 33%. Since the ignition temperature of natural gas is high and it is difficult to be ignited by compression, people combine the dual fuel system by adding a natural gas cylinder, but the two fuel supply systems cannot be used at the same time. In addition, the natural gas gasoline dual fuel system has another disadvantage. The octane number of natural gas is higher than that of gasoline, but when the engine is filled with natural gas, the vehicle cannot improve the thermal efficiency by increasing the compression ratio, and cannot meet the requirements of gasoline explosion resistance (The higher the octane

number of gasoline, the better the explosion resistance, and then the engine can use a higher compression ratio). To meet these needs, people have designed pure natural gas vehicles.

Diesel and natural gas blends are usually used in heavy vehicles or ships. Compared with diesel engines, dual fuel engines emit less $NO_X$. The natural gas diesel dual fuel engine has two combustion forms: one ignites the ignited diesel spray without mixing ahead, and the other ignites the natural gas mixed in advance. When the former method is adopted, a large amount of thermal $NO_X$ and fast $NO_X$ will be generated. When the latter method is adopted, only a little $NO_X$ is generated under partial load or low load, while under high load, the output of no is greatly increased due to the increase in combustion temperature. Nevertheless, the amount of fuel used for ignition is very small, so using natural gas diesel hybrid fuel will still emit less $NO_X$ than diesel engine; however, due to the slit effect, the mixture of natural gas and air is easy to produce more HC emissions.

In order to compare the characteristics of different alternative fuels more clearly, the properties, advantages and disadvantages of different alternative fuels are summarized in Table 2.

**Table 2.** The properties, advantages and disadvantages of different alternative fuels.

| | Fuel | Density (kg/m$^3$) | Heating Value (MJ/kg) | Sulfur (wt%) | Viscosity (40 °C) (mm$^2$/s) | Cetane Number | Heat of Evaporation (kJ/kg) | Flash Point (°C) | Carbon Content (wt%) | Advantages | Disadvantages | Ref. |
|---|---|---|---|---|---|---|---|---|---|---|---|---|
| Biodiesel | HFO | 990 | 40.8 | 1.3 | <700 (50 °C) | >20 | - | >60 | - | 1. Could direct substitute for conventional fuels 2. Carbon neutral | 1. High cost 2. High NO$_X$ and soot emissions 3. Limited production capacity | [23,44] |
| | ULSD | 840 | 42.5–44.8 | <10 mg/kg | 2.4 | 45–55 | 250–290 | 50–82 | 86.6 | | | [14,17] |
| | WCO | 873.8 | 37.5 | <10 mg/kg | 4.395 | 55.3 | 300 | 182.5 | 77.1 | | | [14] |
| | Jatropha oil | 918.6 | 39.774 | - | 49.93 | 40–45 | - | 240 | - | | | [17,28] |
| | WFO | 916.7 | 38.97 | 0.33 | 42.53 | 56 | - | 327 | 75.03 | | | [22] |
| | HVO | 770–790 | - | - | - | >70 | - | - | - | | | [28] |
| Alcohol | Methanol | 790 | 19.674–19.8 | - | 0.59 | 3–5 | 1110 | 11 | 37.5 | 1. Liquid fuel that enables use of upgraded existing 2. Renewable sources | 1. High cost 2. Absence of bunkering infrastructure 3. High greenhouse gas emissions | [15,17,21] |
| | Ethanol | 790 | 28.6 | - | 1.1 | 6 | - | 13 | - | | | [21] |
| | Butanol | 808 | 33.1 | - | 2.63 | 25 | - | 35 | - | | | [28,64] |
| Waste lubricating oil | | 895–986 | 41.8–43.52 | 0.2 | 3.49 | 56.8 | 360 | 71–244 | 84.76 ± 0.75 | 1. High calorific value 2. Beneficial to lubricating fuel injectors 3. Low cost | 1. Contains metallic impurities 2. High greenhouse gas emissions | [27,31,34,44,47,64] |
| Waste plastic oil | HPDE | 800–920 | 45.4 | - | 2.420–2.52 | - | - | 40–48 | 85.3 | 1. Environmental protection 2. Sufficient raw materials | 1. Increase the emission of NO$_X$ 2. High greenhouse gas emissions 3. Lack of policies and infrastructure | [80] |
| | LDPE | 768–802 | 39.1 | - | 1.650–1.801 | - | - | 50 | 85.3 | | | [81] |
| | PP | 767–800 | 40 | - | 2.72 | - | - | 31–36 | 85.61 | | | [82] |
| | PET | 870–900 | 28.2 | - | - | - | - | - | 92.32 | | | [28,83] |
| | PS | 850–860 | 43 | - | 1.4 (50 °C) | - | - | 28 | 62.10 | | | [84] |
| Natural gas | | 0.78 | 47.57 | 0 | - | 130 | - | - | 74.15 | 1. Mature technology 2. Eliminates SO$_X$ pollution 3. Low cost | 1. High greenhouse gas emissions 2. Mast be stored in insulated tanks | [28,85] |

ULSD: Ultralow sulfur diesel, WFO: Waste frying oil, WCO: Waste cooking oil, HVO: Hydrotreated vegetable oil, HPDE: High density polyethylene, LDPE: low-density polyethylene, PP: polypropylene, PET: poly(ethylene terephthalate), PS: polystyrene.

## 6. Emission Reduction Technology

Facing the above engine emission problems caused by different alternative fuels, it is necessary to deal with different types of alternative fuels through technical means to reduce engine emissions and reduce the impact on SCR catalyst. The following are some solutions to reduce engine emissions by pretreatment of different fuels or optimization and modification of engine exhaust system.

### 6.1. Fuel Optimization

6.1.1. Reduce the Viscosity of Biodiesel

Biodiesel has a high viscosity. Although it can improve the lubricity of components, it is not conducive to the volatility of fuel and will lead to incomplete combustion of fuel, resulting in increased emissions of harmful gases such as PM and HC [86]. Moreover, due to the high viscosity of biodiesel, it will also lead to coking of the fuel injector nozzle. The viscosity of biodiesel is mainly related to the length and saturation of the hydrocarbon chain. The extension of the hydrocarbon chain and the increase in saturation will increase the viscosity of biodiesel [86]. There are many methods to reduce the viscosity of biodiesel, such as transesterification, catalytic cracking, blending, emulsification, microemulsion, Fischer Tropsch, preheating, etc. [58]. Tomi et al. [87] also studied the effect of accelerated oxidation on the properties of biodiesel. The viscosity of biodiesel can be regulated by controlling the oxidation time, which helps to ensure that the viscosity of biodiesel is within an acceptable range, which cannot only provide lubrication for various components, but also not increase engine emissions [88–91]. Karabektas et al. [92] investigated the effect of different preheating temperatures on the viscosity of cotton seed oil and further investigated the effect of preheated cotton seed oil on engine performance. The results showed that preheating reduced the viscosity of cotton seed oil and significantly improved the volatility and combustion characteristics of the fuel. Devan and Mahalaxmi blend paradise oil and eucalyptus oil methyl ester, which could adjust the viscosity of the mixture to keep it within the desired range [93]. Vallinayagam et al. [88] mixed pine oil and kapok methyl ester with different proportions, which significantly reduce the viscosity of the mixed fuel, thereby effectively improving the performance of the fuel. The results showed that the emissions of CO and HC reduced 43.25% and 14.9%, respectively, when B25P75 (Pine oil—75% and KME—25%) was used. Moreover, alcohols can effectively reduce the viscosity of biodiesel [22,40,91]. Babu et al. [22] mixed biodiesel with *n*-pentanol alcohol and diesel. After comparing the mixtures with different proportions, they found that the viscosity of the mixed fuel was significantly reduced after adding *n*-pentanol alcohol and diesel, and found that when the ratio of the mixed fuel was biodiesel: diesel: *n*-pentanol (85:5:10), the emissions of CO and HC were significantly reduced.

6.1.2. Excess Hydrogen Method

Although natural gas is a potential alternative fuel, it is a greenhouse gas, and its global warming potential is about 30 to 85 times that of carbon dioxide [73]. In addition, the carbon dioxide emitted by ships accounts for about 2.5~3.5% of the world [75]. The Figure 6 below shows the proportion of carbon dioxide emissions from various ships in 2015.

To solve this problem, scientists have tried to add hydrogen to compressed natural gas. It was found that with the increase of hydrogen content in natural gas fuel, the content of carbon dioxide in emissions decreases rapidly. Liu's further experiments show that using hydrogen mixed natural gas as fuel can also reduce HC and CO emissions [94].

6.1.3. Transesterification

Because the physical and chemical properties of biodiesel prepared by transesterification are similar to commercial diesel, transesterification is the preferred biodiesel preparation method in the world [95,96]. Bilgin et al. [96] studied the effects of NaOH concentration, reaction temperature and reaction time on the kinematic viscosity of methyl ester synthesized by transesterification. The optimum reaction conditions are determined

through comparative experiments, which is conducive to ensure that the fuel viscosity is in an appropriate range and make the fuel burn fully and reduce the influence of PM and HC on SCR catalyst.

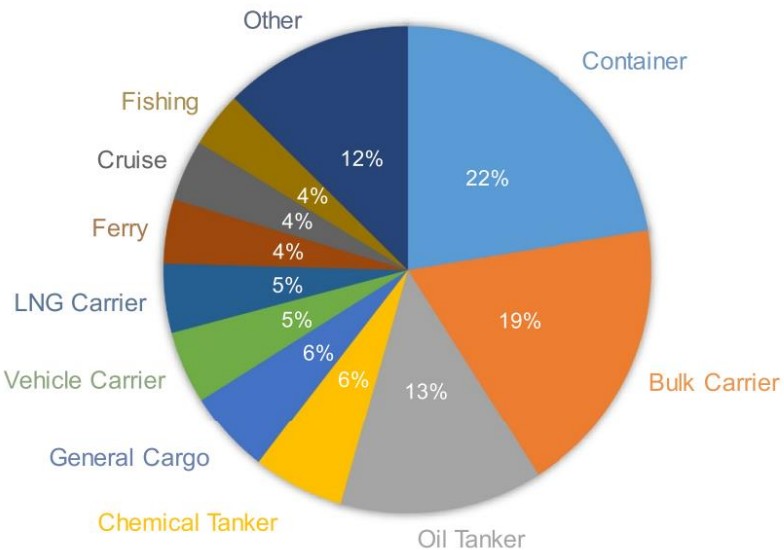

**Figure 6.** The proportion of carbon dioxide emissions from various ships in 2015. Reprinted with permission from [75]. Copyright © 2022, Elsevier B.V.

6.1.4. Microemulsion Technology

Microemulsion refers to the optical isotropic dispersion of oil and water, named by Schulman [97]. The further study of microemulsion shows that the microemulsion is a transparent, thermodynamically stable colloidal dispersion of polar and non-polar phases, stabilized by surfactant, and its particle size is less than 1/4 of visible wavelength [98]. The structure of formed microemulsions is determined by the ratio of oil to water and the nature of surfactant [58]. Compared with hydrotreating vegetable oil, Fischer Tropsch diesel and other alternative fuels such as biodiesel, microemulsion technology has more advantages. Because the moisture in microemulsion increases the latent heat of vaporization of fuel, the maximum temperature inside the cylinder will be reduced, thus reducing the emission of $NO_X$. Kayali et al. [99] found that compared with commercial diesel, diesel oil microemulsion with $C_{14}E_3$ as surfactant and ethanol as surfactant could reduce $NO_X$ emissions by nearly 80% when solar term opening was 30%. Further studies found that this is mainly due to the presence of more water and ethanol in the microemulsion, which makes the fuel have higher latent heat of vaporization, thereby reducing the $NO_X$ emissions of the engine, and the effect of water is much larger than that of ethanol. Microemulsion fuel will also reduce the emission of HC and CO. Although the low combustion temperature of microemulsion will cause incomplete combustion of fuel on the wall, which could increase the formation of HC, the microemulsion fuel contains a large number of oxygen elements, which makes the fuel burned fully, thus reducing the emission of HC and CO. In microemulsion fuel, oxygen exists in water, alcohol, or biodiesel. Oxygen will improve the oxidation of PM particles and soot and reduce the emission of PM and soot.

6.1.5. Catalytic Cracking Method

Catalytic cracking is a method of breaking long-chain molecules to produce small molecules at a certain temperature. Generally, vegetable oil, animal oil and fatty acid methyl ester can be catalytically cracked [100]. It has been reported that bentonite as a catalyst can significantly reduce the viscosity of bio oil prepared from woody biomass [101]. Yakup et al. [102] used bentonite as a catalyst for catalytic cracking of pyrolysis products of almond shell. It was found that the viscosity of almond shell pyrolysis solution decreased significantly by catalytic cracking of bentonite. When the viscosity of biodiesel prepared is

within an acceptable range, it will contribute to fuel atomization and full combustion, so as to reduce the toxic effects of PM, HC and other substances on SCR catalyst.

### 6.1.6. Adjust the Injection Timing

Fuel injection timing has a significant impact on engine emissions [38,103]. In recent years, more and more studies have shown that engine pollutant emissions can be effectively reduced by adjusting engine fuel injection timing [36,38,70,103–106].

Wei et al. [53] found that the combustion temperature of waste cooking oil biodiesel could be effectively increased by optimizing the fuel injection time, which could accelerate the oxidation of soot and reduce the emission of soot. Damodharan et al. [38] found that $NO_X$ emissions from engines decreased as the injection timing was delayed when using waste plastic oil as fuel. This was attributed to a reduction in maximum cylinder temperature and expansion losses, caused by the delay of injection, resulting in a reduction in $NO_X$ emissions. On the contrary, at early injection time, the emissions of smoke, HC and CO would be decreased. This is attributed to the advancing of injection timing, which could increase the maximum cylinder temperature, thereby promoting the oxidation of HC, CO and smoke. Gabiña et al. [105] used waste lubricating oil as an alternative fuel for marine diesel engines and found similar results. Moreover, there are reports that injection timing could effectively solve the problems, which associated with the use of natural gas in these engines [20]. Li et al. [107] found that retarded injection timing would promote the reduction of $NO_X$ emissions from natural gas fueled engines.

Therefore, by summarizing the research results, it is found that the injection timing could reduce the pollutant emission of the engine; however, it can be found that the emission law of $NO_X$ is opposite to the emission law of HC, CO and soot, so it is not only necessary to adjust the injection timing of the engine but also combine with other methods to deal with the engine pollutants.

### 6.2. After Treatment Technology

Only through the engine internal treatment and fuel optimization cannot meet the requirements of the existing emission regulations. Therefore, it is an inevitable way to reduce engine emissions through exhaust gas post-treatment system.

### 6.2.1. DOC Technology

Aslan et al. [108] studied the combination of diesel oxidation catalytic (DOC) and SCR post-treatment system. DOC can effectively reduce the emission of reduced gaseous pollutants from the engine, and the schematic of DOC was shown in Figure 7. It can efficiently oxidize HC, CO and soluble organic component. It can also oxidize NO to $NO_2$, which is conducive to the rapid reduction reaction of SCR [109]. It is found that DOC system can effectively remove CO, HC, NO and soot emissions from engine exhaust, and SCR device behind DOC device can effectively reduce $NO_X$ emissions. DOC devices can also effectively reduce the working load of SCR catalyst, increase the removal efficiency of $NO_X$ and reduce engine emissions.

### 6.2.2. Low Temperature Plasma Technology

Chen et al. [110] improved the removal efficiency of $NO_X$ from ship exhaust by combining low-temperature plasma technology and photocatalytic technology. Low temperature plasma is mainly produced by gas discharge. Under the action of external electric field, dielectric discharge produces a large number of high-energy electrons, which could separate, ionize and excite pollutant molecules, and then carry out a series of physical and chemical reaction processes. Finally, the pollutant molecules become non-toxic and harmless substances [111]. By studying the characteristics of the two technologies, it is found that the key steps are: firstly, let the tail gas pass through the low-temperature plasma reactor to reduce the NO concentration in the tail gas to a lower value. Then, the tail gas is introduced into the photocatalytic reaction device, which can improve the $NO_X$

removal efficiency of the photocatalytic reaction device. The experimental results show that when the two technologies are combined, the no removal efficiency is 65–75%. The removal effect of NO is better than that of one of low-temperature plasma technology and photocatalytic technology. Low temperature plasma technology can convert NO into $NO_2$, which is conducive to rapid reduction reaction of SCR system [112], and helps to reduce $NO_X$ emission.

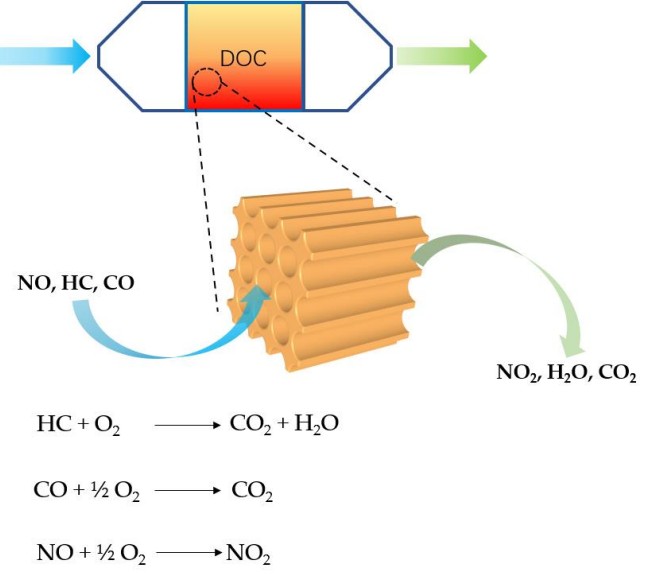

**Figure 7.** The schematic diagram of DOC.

### 6.2.3. DPF Technology

Diesel particulate filters can effectively purify particles in diesel engine exhaust, and the removal rate of soot in emissions can reach 60–90%. It is one of the most effective methods to purify diesel particulate matter [113]. At present, the discussion of DPF technology usually focuses on capture and regeneration.

The interception mechanism of DPF includes direct interception, inertial collision and Brownian diffusion capture. The effect of direct interception is weak in the initial stage, but with the adsorption of particles, the pore will shrink, which will make the later capture effect more superior. For particles with large mass, its trajectory is not easy to change and is easy to be intercepted and collected by inertia. For particles with small particle size, their Brownian motion is obvious at high temperature, which is easy to be obtained through Brownian capture mechanism [114]. In the process of physical adsorption, solid particles are continuously deposited on the filter surface. When the particles reach a certain amount, the exhaust resistance of gas increases, and the exhaust gas will be difficult to be eliminated, which will affect the adsorption efficiency and engine performance. Therefore, in order to maintain the high efficiency of DPF technology, we need to timely clean up the particles deposited in DPF. This process is called DPF regeneration [113,115], and the structure and the regeneration process of DPF or DPF + SCR were shown in Figure 8 [116,117].

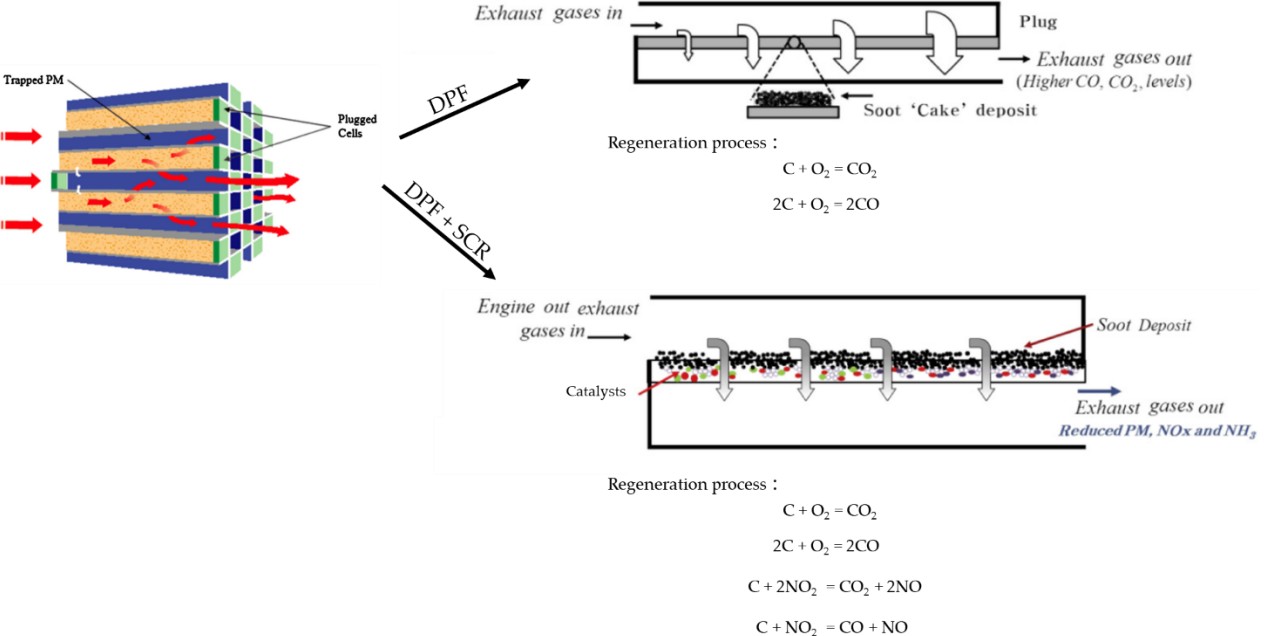

**Figure 8.** The structure and the regeneration process of DPF or DPF + SCR. Reprinted with permission from [116,117]. Copyright © 2022, SAE International, Copyright © 2022, Elsevier B.V.

DPF regeneration is mainly achieved through the oxidation of particles, and the key influencing factors are high temperature, oxygen enrichment and oxidation time. Generally, the temperature required for particulate oxidation is about 500–550 °C, but the exhaust temperature of most diesel engines cannot meet this requirement, so active regeneration or passive regeneration technology is required at this time.

Active regeneration refers to heating DPF to make the temperature reach the ignition point of particles. At present, the main means include fuel injection combustion supporting regeneration, electric heating regeneration, microwave heating regeneration and infrared heating regeneration.

Passive regeneration mainly reduces the ignition point of particles by adding catalyst, so that the combustion reaction can be carried out. Common passive regeneration methods include fuel additive catalytic regeneration filter system, CRT system and CCRT system.

## 7. Summary and Prospect

According to the influence law of different alternative fuels on engine emissions, the following conclusions and prospects are drawn:

1.  Diesel alternative fuel based on waste oil was prepared by a series of purification processes of waste plastic oil or waste lubricating oil. The effect of alternative fuel prepared from waste oil on engine emission was studied through the actual test of engine. It is found that the use of waste plastic oil will reduce the content of PM, which could reduce the negative impact on SCR catalyst [36]; however, the waste plastic oil as an alternative fuel for diesel engines will increase the emission of $NO_X$, which is due to the high heat release rate and high combustion temperature. In contrast, the alternative fuel prepared from waste lubricating oil will increase the emissions of PM and HC, and it was mainly due to the incomplete combustion of the fuel, which cause by the high viscosity of the waste lubricating oil; however, the incomplete combustion of fuel could reduce the combustion temperature, which is best for the decrease of $NO_X$ emission. Controlling the viscosity of alternative fuels within a reasonable range can effectively improve the utilization of alternative fuels, which will be a hot issue in recent years.

2. Biodiesel is a diesel alternative fuel prepared by transesterification, pyrolysis and microemulsion of vegetable oil, animal fat and waste edible oil. The sulfur content in biodiesel is very low, which can effectively reduce the concentration of $SO_2$ in exhaust gas. Biodiesel is rich in oxygen-containing functional groups, which can effectively improve fuel utilization and reduce PM and HC emissions; but the biodiesel will increase the emissions of $NO_X$, which due to the high combustion temperature. The biodiesel can effectively reduce the adverse effect of $SO_2$ on SCR catalyst and the blockage of SCR catalyst by PM and HC. Biodiesel is a potential alternative fuel to diesel.

3. Alcohol fuel is usually mixed with biodiesel and commercial diesel to prepare engine alternative fuel. It is found that higher alcohols have good blending ability and cetane number, so it is an excellent fuel improver. The addition of higher alcohol will effectively reduce the $NO_X$ emission of the engine and reduce the working load of SCR catalyst; however, with the increase of alcohol concentration, HC emission will also increase. Therefore, in the future research, we should systematically study the content of alcohols to understand the impact of different alcohols on engine emission law at different concentrations.

4. Natural gas is an efficient and clean energy with wide distribution and rich reserves. Since it does not contain sulfur elements, it can effectively reduce $SO_2$ emissions. Moreover, the use of natural gas can also effectively reduce the content of PM in exhaust gas. In addition, the lean combustion of natural gas engines will also reduce the content of $NO_X$ in emissions.

5. The use of technical means to improve fuel characteristics or to treat exhaust gas can also effectively solve the pollution problem of emissions. We can achieve the effect of reducing emissions by improving the physical properties of the fuel, such as reducing the viscosity, or improving the chemical properties, such as adding excess hydrogen. The use of DOC, DPF, SCR and other technologies to treat the exhaust gas can also greatly reduce the pollutant content in the discharge.

**Author Contributions:** S.F.: Writing. S.X.: Figures, Study design. P.Y.: Data collection, Writing. Y.X.: Data collection, Data analysis. B.S.: Supervision, Writing. Z.L.: Literature search. C.Z.: Data collection. X.W.: Data analysis. Z.W.: Study design. J.M.: Writing. W.K.: Data interpretation. All authors have read and agreed to the published version of the manuscript.

**Funding:** The project was supported by National Key Research and Development Program of China (2018YFB0605101), Key Project Natural Science Foundation of Tianjin (18JCZDJC39800), Key Research and Development Program of Tianjin (19ZXSZSN00050, 19ZXSZSN00070).

**Data Availability Statement:** Not applicable.

**Conflicts of Interest:** The authors declare no conflict of interest.

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
