# Peer review of "The Impact of Alternative Fuels on Ship Engine Emissions and Aftertreatment Systems: A Review"

_catalysts, doi:10.3390/catal12020138_

Round 1
Reviewer 1 Report
Dear Authors,
Emissions of internal combustion engines are a very topical issue. The article describes biofuels and their impact, as well as equipment for exhaust after-treatment. In my opinnion, if the article will serve as a reference for the study, it is necessary to describe in detail the individual chemical reactions of emissions production, as well as a detailed explanation of the reactions in the exhaust after-treatment systems. (Catalysis, oxidation on so on)
Best regards
Reviewer 2 Report
Manuscript Number: catalysts-1559075
The authors have tried to summarize the influence of different alternative fuels on marine engines, such as waste oils (section 2 with only 2 types of waste oil), biodiesel derived from vegetable oil and waste edible oil (section 3), alcohol fuels (section 4), and natural gas (section 5). They have also added some methods used to reduce the emissions of marine engines. The introduction does not include any comparison with other reviews in the field. Authors should put some comparison with other reviews on the topic and explain how this review is different and why it is significant. There are exclusive reviews on emission reduction technology of marine engines(10.1016/j.scitotenv.2020.144319,10.1007/s11630-020-1342-y, 10.4236/ojms.2019.93012 ) and on alternative fuels for marine engine, 10.1177/1475090214522778, etc. Overall, the review is not comprehensive enough for publication at its current state. Some of the suggestions and comments are listed below:
- The author should mention about the characteristics (high energy density, GHG neutral, availability +security of supply), that any alternative fuel must meet as marine fuel, before introducing the alternative fuels. They should include some information on how the marine fuel quality is regulated. Cite "Emissions from ships." Science5339 (1997): 823-824.
- Authors mentioned in line 61-64, “Marine engines have strong applicability to different alternative fuels and can be used directly without modification for alternative fuels mixed in different forms.” This statement should be backed up with references and proper justification.
- The statement in line 74, “there is a shortage of fossil fuels”, should be backed up with some reference.
- Figure 1 is missing reference.
- Authors have introduced section 2, highlighting the importance of recycling waste oil as an alternative fuel for diesel. But they have not mentioned how they are comparing different alternative fuels. It would be difficult for readers to correlate how the comparison is done and to understand which one is better than the other. It seems authors have just picked up some studies on each of the 5 alternative fuels and briefly discussed how it affects the emission, rather than highlighting the advantages and limitations of each. They have mentioned cost of biodiesel is higher than diesel but they have not mentioned anything about cost of waste plastic oil or waste lubricating oil.
- Status of waste lubrication oil in countries other than China has not been mentioned.
- The authors should also consider the benefits and disadvantages of adopting the injection timing adjustment strategy to address certain engine emission problems. Cite Mar. Sci. Eng.2021, 9(10), 1072; https://doi.org/10.3390/jmse9101072. The simulation results showed reductions in the in-cylinder peak pressure and temperatures, as well as the emission formations, in the DF modes in comparison to the diesel mode. The DF mode could significantly reduce nitric oxide (NO) emissions (up to 96.225%) of DME compared to the diesel mode.
- Line 521 needs reference.
- In section 7, for summary, point 1- Authors did not mention the effect of either WPO or WLO on the NOx reduction. Same with point 2, effect of biodiesel on NOx is not mentioned in the summary.
- Line 182-grammatical error
- Line 395- suggest some literature which has reported reducing viscosity of biodiesel
- Line 422 -typo.
Round 2
Reviewer 2 Report
Authors have done a sincere work to address the comments.